# Loss of p16: A Bouncer of the Immunological Surveillance?

**DOI:** 10.3390/life11040309

**Published:** 2021-04-02

**Authors:** Kelly E. Leon, Naveen Kumar Tangudu, Katherine M. Aird, Raquel Buj

**Affiliations:** 1UPMC Hillman Cancer Center, Department of Pharmacology & Chemical Biology, University of Pittsburgh School of Medicine, Pittsburgh, PA 15213, USA; kel170@pitt.edu (K.E.L.); tangudunk@upmc.edu (N.K.T.); katherine.aird@pitt.edu (K.M.A.); 2Biomedical Sciences Graduate Program, Penn State College of Medicine, Hershey, PA 15213, USA

**Keywords:** senescence-associated secretory phenotype (SASP), senescence, cell-cycle, melanoma, pancreatic adenocarcinoma, tumor infiltration, chemotherapy resistance

## Abstract

p16^INK4A^ (hereafter called p16) is an important tumor suppressor protein frequently suppressed in human cancer and highly upregulated in many types of senescence. Although its role as a cell cycle regulator is very well delineated, little is known about its other non-cell cycle-related roles. Importantly, recent correlative studies suggest that p16 may be a regulator of tissue immunological surveillance through the transcriptional regulation of different chemokines, interleukins and other factors secreted as part of the senescence-associated secretory phenotype (SASP). Here, we summarize the current evidence supporting the hypothesis that p16 is a regulator of tumor immunity.

## 1. Introduction

Since the 19th century, immunologists have been speculating about the idea that the immune system may be a strong, efficient, and specific weapon against cancer (reviewed in [1]). However, it was not until the beginning of the 21st century that scientists began to understand the mechanisms behind tumor immunity and to develop immunotherapy regimens [2]. In parallel, the senescence field discovered that senescent cells, although in a stable state of cell cycle arrest [3], are highly active and acquire a pro-inflammatory microenvironment termed the senescence-associated secretory phenotype (SASP) [4,5]. Through the SASP, senescent cells can modify their microenvironment and regulate other cells, including cells of the immune system [5,6,7,8,9,10]. 

Among the multiple pathways that are commonly deregulated in cancer and senescence [11], the p16^INK4A^ (hereafter called p16) pathway is particularly intriguing. On one hand, loss of p16 is a common feature of cancer that causes an increase in the proliferative capacity of the cell [12]; on the other hand, upregulation of p16 is a hallmark of senescence that contributes to the characteristic state of cell cycle arrest [13]. Interestingly, recent publications demonstrate that suppression of p16 correlates with decreased activity of immune cells [14,15,16,17], and our recent publication shows that p16 suppression decreases expression of the SASP [18]. Altogether these data suggest that p16 may have a, yet unknown, role in the regulation of tumor immunity that might have important implications in the treatment of cancers with low or null p16 expression. In this review, we describe the roles of p16 and the SASP in both senescence and cancer and dissect the latest publications that support the hypothesis that p16 may be a regulator of tumor immunity.

## 2. p16 in Cancer and Senescence

Cyclin Dependent Kinase Inhibitor 2A (*CDKN2A*) located in chromosome 9p21 is a tumor suppressor gene that encodes p14^ARF^ (p19^ARF^ in mice, hereafter p14 and p19) and p16 proteins using two different open reading frames. While p14^ARF^ is involved in the regulation of the p53 pathway (reviewed in [19]), p16′s canonical role is to inhibit the assembly and activation of the cyclin-dependent kinases CDK4/6, impairing the hyperphosphorylation of the retinoblastoma (RB) protein and the E2F-mediated expression of proliferation-promoting genes [20]. Due to its role as cell cycle brake, it is not surprising that ~50% of human cancer shows decreased expression of *CDKN2A* [12]. Interestingly, although the most common alterations of p16 are deletions and promoter hypermethylation affecting both p16 and p14 [21,22,23], cancer-associated mutations are more commonly found in p16 than p14 [24], suggesting a critical regulatory role of p16 in the cell. Importantly, loss of p16 alone is not enough to produce cancer, mainly because normal cells have other mechanisms to abrogate the cell cycle progression (e.g., p53, CHK1/2, APC) (reviewed in [25]). However, it has been shown that the suppression of p16 facilitates malignant transformation of cells upon different hyperproliferative signals and stressors such as oncogenes, oxidative stress, ionizing radiation, and others. All together these data suggest that tight regulation of p16 expression is critical to maintain healthy cellular proliferation. 

On the other hand, p16 is known to be highly expressed during cellular senescence [26,27,28]. Senescence is a stable state of cell cycle arrest acquired upon different stressors such as aberrant proliferative signals or DNA damage among others [29]. Since cells are continuously affected by a wide variety of stressors, it is not surprising that a wide variety of senescence inducers exist, including: oncogenes, ionizing radiation, genotoxic chemicals, reactive oxygen species, chemotherapeutic agents or shortage in dNTPs among others (reviewed in [30]). Interestingly, in the vast majority of cases, the distinctive cell cycle arrest is achieved by upregulation of the p16 protein as a direct consequence of the pathways (e.g., p38 and ERK) [31,32,33,34], epigenetic factors (e.g., polycomb) [35,36,37] and transcription factors (e.g., ETS2 and AP-1) [38,39,40,41] altered by the senescence inducers. In this regard, several groups including our laboratory have shown that suppression of p16 can bypass senescence [42,43,44,45,46], indicating that p16 is critical to maintain the senescence phenotype. 

In addition to its canonical role regulating the cell cycle, an increasing amount of evidence indicates that p16 has non-canonical, RB-independent roles. For instance, our laboratory found that suppression of p16 bypasses oncogene-induced senescence in part by promoting an increase in nucleotide and deoxyribonucleotide levels in a mechanism mediated by the mTORC1 complex and in an RB-independent manner (i.e., operating outside the cell cycle) [42]. Additionally, p16 has been found to regulate tumor suppressive miRNAs, mitochondria biogenesis, oxidative stress, transcription factors such as AP-1 and NF-κB or protein translation though EEF1A2 (reviewed in [47]). Altogether this shows that p16 is not a simple cell cycle brake, but also a regulator of other processes. Indeed, the observation that the large majority of cancer-associated mutations targeting *CDKN2A* are mainly affecting the p16 open reading frame, reinforces the importance of p16 as a regulator of cellular physiology and stresses the necessity to further investigate p16-mediated regulatory processes.

## 3. Role of SASP in Senescence and Cancer

Together with the signature cell cycle arrest, the SASP is one of the most prominent phenotypes of senescent cells. The SASP is composed of various soluble and non-soluble factors including cytokines, chemokines, and proteases that are highly expressed and secreted by senescent cells, creating a pro-inflammatory microenvironment that affects themselves and other non-senescent cells in an autocrine and paracrine fashion, respectively [48,49]. In part because of its ability to modify the environment and impact the behavior of other cells, the SASP is tightly regulated at multiple levels. Transcriptionally, various factors (C/EBPβ and NF-κβ) [9,10], upstream regulators (p38, MAPK, GAT4A, p53, or ATM) [5,50,51,52,53] and non-coding RNAs (miRNAs, lncRNAs and circRNAs) [54] have been described to regulate SASP expression. Additionally, mTORC1-mediated translational regulation of MAPKAPK2 and IL1A has been shown to impact several SASP factors [55,56]. Finally, during senescence, there is a rearrangement of the genomic architecture leading to a new TAD (topologically associated domain) landscape [57,58]. This rearrangement is induced by changes in DNA methylation, nucleosome organization and histone modification, giving rise to the so-called senescence-associated heterochromatin foci (SAHF) [59,60,61,62,63,64]. There is a close relationship between the SAHF and the SASP and multiple epigenetic modifiers such as HMGB2, BRD4, MLL1, macroH2A1 or SIRT1, among others, have been shown to impact expression of the SASP [65,66,67,68,69]. In this regard, recent work from our laboratory shows that increased expression of the histone methyltransferase DOT1L upon oncogene-induced senescence, drives the expression of the major SASP-inducer IL1A though increased deposition of the active histone marks H3K79me2/3 at the *IL1A* gene loci [70]. 

Due to its inherent arrest of the cell-cycle described above, cellular senescence has been considered a bona fide tumor suppressor mechanism [71,72]. However, in the late 90s and early 2000s new experiments demonstrated that senescent cells can promote cellular proliferation and tumor growth due to acquisition of the SASP [7,73,74,75]. The SASP is highly dynamic and variable depending on several factors such as genetic background, cell type, the inducer of senescence, and the time of which senescence has occurred [76,77,78,79]. Due to its complex and variable nature, physiological roles of the SASP have not been well delineated. On one side, the SASP has been shown to contribute to the immunological surveillance, i.e., the process whereby the cells of the innate and adaptative immune system detect and destroy damaged cells [80,81,82]. For example, SASP secretion by senescence hepatocytes promotes T-cell mediated immunological surveillance within the liver, promoting the clearance of pre-malignant senescent cells and hence avoiding tumor progression [83]. On the other hand, the SASP has also been shown to contribute to numerous detrimental effects such as tumor promotion and progression and therapy resistance (reviewed in [84]). For instance, it has been described that some SASP secreted by senescent stromal cells promote an immunosuppressive microenvironment increasing the number of myeloid-derived suppressors cells (MDSCs), thus impairing immunological surveillance and promoting tumor growth [85,86]. In this regard, the dynamic nature of the SASP and its time-dependent regulation seems to be key to understanding the interaction with the immune system and hence the positive or negative outcome in the tumoral area [76,87]. Additionally, the SASP is not exclusive to senescent cells, and inflammatory phenotypes similar to the SASP, known as “SASP-like”, have been described in tumor cells [67,88]. Indeed our laboratory has found that similar to senescent cells, different tumor types display different SASP (SASP-like) profiles [18], further demonstrating the wide variability of the SASP. Whether this indicates that those tumors bypassed senescence at some point during their malignant transformation is still unknown and more research is needed. However, it is becoming clearer that both the senescent cells in the tumor environment and the tumor cells themselves contribute to the maintenance of a SASP-driven inflammatory microenvironment. Therefore, it is imperative to delineate the mechanisms of SASP expression and secretion as well as map the SASP composition upon different conditions, tissue types and timeframes to design efficient and personalized immunotherapies.

## 4. p16 Regulation of Tumor Immunity

We previously described that canonical and non-canonical roles of p16 in part regulate cellular homeostasis. However, is it possible that p16 regulates other processes that impact the cellular microenvironment? Recent evidence demonstrates that loss of *CDKN2A* expression in tumor cells correlates with different immunological processes within the tumor that may impair immunological surveillance suggesting that p16 not only regulates cellular homeostasis but also tissue homeostasis. Below we will dissect the current evidence that implicates p16 as a regulator of intratumor immunity.

The first piece of evidence was published by Balli et al., [14]. These authors used a previously published expression signature based on two key cytolytic effectors (*GZMA* and *PRF1*) upregulated upon CD8+ T cell activation [89] to assess the intratumoral cytolytic T-cell activity in pancreatic adenocarcinoma samples from The Cancer Genome Atlas (TCGA). Interestingly, non-silent mutations and deletions of *CDKN2A* correlated with decreased cytolytic activity. One year later, Wartenberg et al. [15] investigated the immune cell composition within the microenvironment in a series of pancreatic ductal adenocarcinoma and cross compared it with a high-throughput analysis of somatic mutations. They found that high mutation rates of *CDKN2A* correlate with a so-called immune-escape microenvironment, which is a microenvironment poor in T and B cells and enriched in FOXP3+ Tregs. Consistent with this result, Morrison et al. [16] found that loss of *CDKN2A* significantly correlates with immune deserts, defined by a profile of 394 immune transcripts. These pieces of evidence suggest that low *CDKN2A* expression both impacts the number and the activity of the intratumoral immune cells. Moreover, suppression of *CDKN2A* in mesenchymal stromal cells has been shown to decrease CD11b+ Gr-1^hi^ neutrophils, CD11b+ Gr1^low^ monocytes and CD45-CD31-Integrinα7+ satellite cells in a model of chronic inflammatory myopathy [17]. Although all those investigations are based on correlative analysis, these data may indicate that *CDKN2A* is necessary for regulation of the physiological immune response upon different inflammatory events. However, it should be noted that there are some publications that disagree with this thesis [90,91,92]. Thus, further mechanistic studies are necessary to determine in which circumstances suppression of *CDKN2A* in non-immune cells decreases immunological surveillance.

How does the suppression of p16 abrogate immunological surveillance? A recent publication from our laboratory demonstrates that the specific knockdown of p16 in oncogene-induced senescent cells leads to decreased expression of several SASP genes, including the most well characterized cytokines *IL6*, *CXCL8* and *CSF3*, the proteases *MMP3*, *PLAU* and *PLAT*, the growth factors *AREG*, *EREG* and *VEGFA* and the glycoprotein *ICAM1* [18]. Early suppression of p16 bypasses senescence in vitro and in vivo [18,42,43,93,94]; however, we demonstrated that knockdown of p16 at late time points upon oncogene-induced senescence does not bypass senescence but still decreases the expression of *IL6* and *CXCL8* [18], suggesting that this is uncoupled from the senescence-associated cell cycle arrest. Additionally, we found that low *CDKN2A* expression in tumors of 6 different types, including melanoma and pancreatic adenocarcinoma, correlated with a decreased SASP signature [18], further demonstrating the role of p16 in regulation of the SASP. Consistent with our observation, other articles have found decreased expression of SASP factors upon suppression of p16 in a murine model of intervertebral disc regeneration [95] and liver fibrosis [96]. Interestingly, induction of senescence through p16 overexpression does not increase SASP gene expression [50], suggesting that p16 is necessary but not sufficient to induce the SASP. As we have previously discussed, the SASP has pleiotropic and context-dependent effects that can both promote tumor progression and enhance anti-tumor immunity (Reviewed in [97]); thus, it is plausible that the observed decreased of immunological surveillance in tumors with suppression of *CDKN2A* is mediated by decreased SASP. More studies are needed to understand the exact mechanism whereby suppression of p16 leads to decreased SASP gene expression and whether this leads to decreased immunological surveillance and tumor growth. Additionally, in those tumors where loss of p16 occurs due to deletion of the chromosome 9p21 locus, adjacent genes such as *MTAP* and the interferon α and β cluster may be also affected and lost [98,99]. Previous studies have suggested that melanomas with low *MTAP* expression have decreased cGAS-STING signaling [100], a pathway strongly involved in SASP expression though NF-κβ regulation [101,102,103]. Additionally, codeletion of *CDKN2A* and the interferon α and β cluster has been linked with decreased expression of immune cell genes in melanoma tumors [104]. Therefore, it is likely that multiple mechanisms exist in tumors with loss of 9p21 to suppress SASP gene expression and to modulate the tumor microenvironment.

How is immunological surveillance initiated on the tumor microenvironment? A large amount of evidence suggests that intratumoral senescence induction is critical for activation of the immune system and clearance of cancer cells [83,86,105,106], and abrogation of senescence-inducing pathways, mainly p16 and p21, have been shown to be critical to promote immune-checkpoint inhibitors resistance [107,108]. Thus, which is the main requirement for intratumoral immune system activation: senescence induction or high p16 expression? This is a complicated question since most senescent cells upregulate p16 as a manner to maintain the cell cycle arrest [26,109], and early abrogation of p16 activity overcomes senescence in vitro and in vivo [42,43,93,94], thus likely reducing the intratumoral senescence burden. In this regard, our previous data demonstrate that although etoposide can induce senescence in melanoma cells with stable p16 knockdown, these cells fail to increase the expression of *IL6* and *CXCL8* [18]. Moreover, although tumors with low p16 expression show a significant decrease in SASP factors, we did not observe differences in the amount of intratumoral senescent cells [18]. Interestingly, Novais et al. also found that suppression of p16 in a model of intervertebral disc regeneration decreases the SASP without altering the onset of senescence [110]. Altogether, these data suggest that p16 may regulate the tumor microenvironment and by extension intratumor immunity independently of senescence induction. This indeed, may explain previous observations where overexpression of p16 induces a cell cycle arrest without the SASP [50].

Importantly, the lack of SASP expression observed in cells with p16 suppression indicates that induction of senescence as a mechanism to abrogate malignant proliferation may be a suitable and safe therapy for cancers with null or very low p16 expression. More experiments analyzing the immune response landscape as well as the mechanisms by which p16 decreases SASP are needed to understand this observation and to develop better treatments for the ~50% of human cancers with decreased p16 expression [12].

## 5. Conclusions

In conclusion, suppression of p16 in tumor cells decreases the expression of interleukins, chemokines and other factors belonging to the SASP that in turn may remodel the tumor microenvironment, thereby impairing immunological surveillance (Figure 1). The observation that suppression of p16 decreases SASP gene expression is sustained by different studies in multiple models including cancer and fibrosis. Additionally, there are multiple high-throughput correlative studies in different diseases suggesting that low *CDKN2A* activity correlates with a decreased number and activity of intratumoral immune cells. Altogether, this suggests that p16 suppression is not only a cell cycle regulator but also a regulator of tissue homeostasis. More research is needed to understand whether this is a direct or indirect effect and whether this is due to canonical or non-canonical p16 roles. This is imperative since currently there are not pharmacological treatments specifically for p16-non canonical pathways [47].

## Figures and Tables

**Figure 1 life-11-00309-f001:**
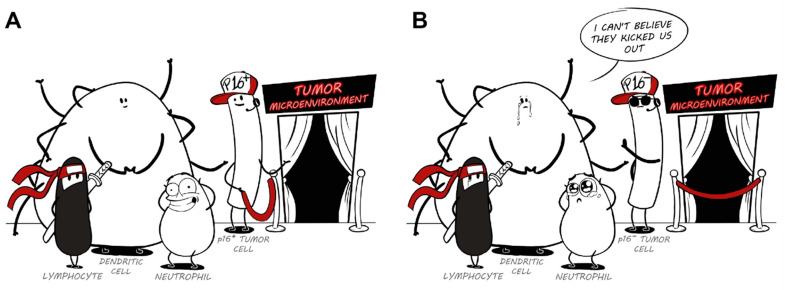
Cartoon representing the hypothesis discussed in this review: the possible role of p16 in regulation of immunological surveillance. (**A**) Represents a p16 positive tumor where cells of the immune system are invited into the tumor microenvironment and hence tumor immunosurveillance is fostered. (**B**) Represents a p16-null tumor, which correlates with decreased number and activity of immune cells, thus impairing immunosurveillance and promoting tumor growth.

## Data Availability

Not applicable.

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
