# Peer review of "Loss of p16: A Bouncer of the Immunological Surveillance?"

_life, 2021, doi:10.3390/life11040309_

Round 1

Reviewer 1 Report

The authors comprehensively detailed and discussed the non-cell autonomous role of p16 in immune surveillance in this review.

There a just a few additional points I am hoping the authors can address: 

  1. The authors cite their own paper (ref. 18), where they show that IL-6 and CXCL8 levels are reduced following p16 suppression in the context of senescence. Were these the only SASP factors looked at? Have other publications looked at SASP expression directly following p16 suppression? Since IL-6 and IL-8 are known to how pro-tumorigenic and anti-inflammatory roles, the authors should comment/speculate on whether p16-mediated SASP regulation may block rather than promote anti-tumor immunity, and whether p16 may have SASP independent roles in immune surveillance.
  2. The interferon cluster (9p21.3) is located adjacent to the CDKN2A/2B locus. While some cancers harbor focal deletions in just CDKN2A, many have larger deletions or copy number losses that encompass not just CDKN2A but also genes within the interferon cluster (for example, see Linsley et al PLOS ONE 2014; Li et al Clinical Cancer Research 2018; Roy et al. Cancer Cell 2016). Given the role of interferons in immune surveillance, it is plausible that some of the immunological effects seen in human cancers with p16 loss may be due in part to loss of interferon signaling in tumors with larger deletions encompassing the interferon cluster. The authors should mention this possibility in their review.
  3. It would be helpful to the reader to have a schematic figure (in addition to the cartoon in Fig. 1) that outlines what it summarized in the text about the role of p16 in cell cycle arrest, senescence, SASP, cancer, and ultimately immune surveillance. 

Author Response

  1. The authors cite their own paper (ref. 18), where they show that IL-6 and CXCL8 levels are reduced following p16 suppression in the context of senescence. Were these the only SASP factors looked at? Have other publications looked at SASP expression directly following p16 suppression? Since IL-6 and IL-8 are known to how pro-tumorigenic and anti-inflammatory roles, the authors should comment/speculate on whether p16-mediated SASP regulation may block rather than promote anti-tumor immunity, and whether p16 may have SASP independent roles in immune surveillance

Response: We thank the reviewer for this comment. In our previous publication (Reference 18 in the revised manuscript), we found that specific knockdown of p16 in a model of BRAFV600E-induced senescence decreased the expression of several SASP genes including IL6, CXCL8, CSF3, MMP3, PLAU, PLAT, AREG, EREG, VEGFA and ICAM1. This information has been included on the revised manuscript lines 181-183. In addition to our publication, decreased expression of SASP factors upon suppression of p16 has been found in a murine model of intervertebral disc regeneration and liver fibrosis (References 95 and 96 in the revised manuscript). This is specified in the revised manuscript lines 190-192. We agree with the reviewer that both IL6 and CXCL8 have pleiotropic and context-dependent effects that can both promote tumor progression but also enhance anti-tumor immunity. This discussion is included in the revised manuscript lines 194-198.

  1. The interferon cluster (9p21.3) is located adjacent to the CDKN2A/2B locus. While some cancers harbor focal deletions in just CDKN2A, many have larger deletions or copy number losses that encompass not just CDKN2A but also genes within the interferon cluster (for example, see Linsley et al PLOS ONE 2014; Li et al Clinical Cancer Research 2018; Roy et al. Cancer Cell 2016). Given the role of interferons in immune surveillance, it is plausible that some of the immunological effects seen in human cancers with p16 loss may be due in part to loss of interferon signaling in tumors with larger deletions encompassing the interferon cluster. The authors should mention this possibility in their review.

Response: We thank the reviewer for this great suggestion. We have added this discussion to our revised manuscript (lines 201-215)

Reviewer 2 Report

General comments

This review is giving a short overview on the cell cycle inhibitor p16INK4a known as a tumor suppressor and involved in senescence, but here linked to tumor immunity. Recent research on p16 and the senescence-associated secretory phenotype (SASP), often of the authors themselves, is discussed in relation to other literature of the field to assess the role of p16 in  immunological surveillance during tumorigenesis. The manuscript summarizes very interesting new aspects on p16, but it needs revision regarding the minor points below.

Minor points

Line 33-36: loss of p16.. ref. 12; upregulation of p16.. ref. 13

Ref. 12 Romagosa et al. deals with overexpression of p16, please correct.

Line 68: Please change font size.

Line 107-111: Reading the text I understand that ref. 55 is from your lab, but it is not. Please reword.

Line 188-189: ..overexpression of ..senescence does not increase.. Senescence is not overexpressed, please correct.

Line 202: Please explain ICI or avoid the abbreviation.

Line 179-180 and 205-206: nearly exact repetition of the phrase, please change.

Figure 1: Labeling of the cells is too small, please enlarge.

References: Order of references, which are accumulatively cited, e.g. (5-10) or (30-40), in the reference list would be nice.

Please correct typos in the whole text and a double “that” (line183).

Author Response

  1. Line 33-36: loss of p16.. ref. 12; upregulation of p16.. ref. 13

Response: We thank the reviewer for this comment. The text has been fixed (lines 33-34 in the revised manuscript).

  1. 12 Romagosa et al. deals with overexpression of p16, please correct. 

Response: We thank the reviewer for commenting on this. We have changed the reference in the revised manuscript.

  1. Line 68: Please change font size. 

Response: The font size has been corrected in the revised manuscript.

  1. Line 107-111: Reading the text I understand that ref. 55 is from your lab, but it is not. Please reword.

Response: We apologize for this misunderstanding. It has been fixed in the revised manuscript (line 110).

  1. Line 188-189: ..overexpression of ..senescence does not increase.. Senescence is not overexpressed, please correct.

Response: We apologize for this misunderstanding. The sentence has been rewritten in the revised manuscript (lines 192-193).

  1. Line 202: Please explain ICI or avoid the abbreviation.

Response: We apologize for the missing explanation. ICI refers to immune-checkpoint inhibitors. This has been corrected in the revised manuscript (line 202).

  1. Line 179-180 and 205-206: nearly exact repetition of the phrase, please change.

Response: We apologize for repeating this phrase. It has been corrected in the revised manuscript (lines 223-224).

  1. Figure 1: Labeling of the cells is too small, please enlarge.

Response: We thank the reviewer for this suggestion. The size of figure labels in Figure 1 has been increased (New Figure 1).

  1. References: Order of references, which are accumulatively cited, e.g. (5-10) or (30-40), in the reference list would be nice.

Response: We thank the reviewer for this suggestion. Accumulative citations have been avoided in the revised manuscript.

  1. Please correct typos in the whole text and a double “that” (line183).

Response: We apologize for this typo. It has been corrected in the revised manuscript (line 185-187).

Reviewer 3 Report

The organization of the manuscript and the concept are good and provide an interesting insight in p16 role in the immune system. There are some minor modifications that you may consider.

-It may be difficult, but a figure delineating the pathways in which p16 protein is involved in immunological response in cancer cells and tumor microenvironment would be useful to the reader.

-Some seminal studies are missing from the review, PMID: 2428504 (presence of p16 in T cell upon antigen activation); PMID: 26840489 (extension of  lifespan by removal of p16); PMID: 25524798 (clinical evidence of p16 loss used to stratify patients and achieve antitumoral efficacy); PMID: 30904769 (increased expression of p16 upon trauma injury in microglia). Please consider to include them if appropriate.

Author Response

  1. It may be difficult, but a figure delineating the pathways in which p16
    protein is involved in immunological response in cancer cells and tumor
    microenvironment would be useful to the reader.

Response: We thank the reviewer for this suggestion. As we stated in the response to reviewer #1 comment 3, we have assessed the possibility of making a schematic delineating p16 pathways and their relationship with the SASP and the immune surveillance. However, figures can be easily misinterpreted out of context. This is particularly critical for this review where all the articles indicating a potential relationship between loss of p16 and decreased immune surveillance are highly correlative, therefore more mechanistic studies are needed. We have stressed this message at multiple places withing the revised manuscript to avoid possible misinterpretation.

  1. Some seminal studies are missing from the review, PMID: 2428504
    (presence of p16 in T cell upon antigen activation); PMID: 26840489
    (extension of  lifespan by removal of p16); PMID: 25524798 (clinical
    evidence of p16 loss used to stratify patients and achieve antitumoral
    efficacy); PMID: 30904769 (increased expression of p16 upon trauma
    injury in microglia). Please consider to include them if appropriate."

Response: We thank the reviewer for this suggestion. Reference PMID 26840489 has been included in the revised manuscript (line 65).